# The Cystic Anechoic Zone of Uterine Cavity Newly Observed during Controlled Ovarian Hyperstimulation Affects Pregnancy Outcomes of Fresh Embryo Transfer

**DOI:** 10.3390/jcm12010134

**Published:** 2022-12-24

**Authors:** Yizheng Tian, Shengrui Zhao, Jianan Lv, Hong Lv, Lei Yan

**Affiliations:** 1Center for Reproductive Medicine, Shandong University, Jinan 250100, China; 2Key Laboratory of Reproductive Endocrinology of Ministry of Education, Shandong University, Jinan 250100, China; 3Shandong Key Laboratory of Reproductive Medicine, Jinan 250012, China; 4Medical Integration and Practice Center, Shandong University, Jinan 250100, China; 5Gynecology Department, Reproductive Hospital Affiliated to Shandong University, Jinan 250001, China

**Keywords:** cystic anechoic zone of uterine cavity, pregnancy outcomes, in vitro fertilization, endometrial receptivity, transvaginal ultrasound

## Abstract

During controlled ovarian hyperstimulation (COH), cystic anechoic zones in the uterine cavity are occasionally visible. This retrospective matched cohort study collected information on patients who underwent in vitro fertilization/intracytoplasmic injection (IVF/ICSI) from January 2014 to December 2020 at our center. The propensity score algorithm matched 179 cases that had uterine cystic anechoic zones, with 358 which did not have uterine cystic anechoic zones cases. After matching, the live birth rate (38.0% vs. 48.6%, *p* = 0.025) of patients with uterine cystic anechoic zones was lower than that in the no uterine cystic anechoic zone group, while for clinical pregnancy miscarriage rate (22.2% vs. 12.4%, *p* = 0.031), the rate was higher. The results showed no correlation in the association between live birth rate (r = −0.027, *p* = 0.718), clinical pregnancy rate (r = −0.037, *p* = 0.620) or biochemical pregnancy rate (r = −0.015, *p* = 0.840) and the diameters of the cystic anechoic zones in the uterine cavity. There was a significant difference in the type of endometrium between the two groups (*p* < 0.001). The result of this study can provide guidance to patients on whether to undergo fresh embryo transfer in the current cycle.

## 1. Introduction

In recent years, emphasis has been placed on the uterine environment in order to achieve higher pregnancy rates in assisted reproductive technology (ART) [1,2]. The endometrium is an important factor in the implantation process [3]. Sometimes, uterine lesions are found during COH. It has been shown in several previous studies that echogenic, smooth, intracavitary masses in the uterus that appeared during COH does not have a harmful effect on pregnancy outcomes [4,5]. On an ultrasound, leiomyomas are usually hypoechoic, discrete, round masses [6]. Submucosal leiomyoma (FIGO 0–2) is strongly associated with low persistent pregnancy rate and miscarriage [7,8,9,10]. Endometrial hyperplasia (EH) can be cystic or asymmetrically thickened, and can mimic other focal entities, such as endometrial polyp (EP), submucosal myomas or cancer [6]. These uterine lesions may affect pregnancy outcomes by disrupting endometrial receptivity [11,12,13].

During COH, cystic anechoic zones in the uterine cavity can be occasionally found. The cystic anechoic zone has a cyst-like structure. Its sonographic features are as follows: one or more anechoic zones detected in the uterine cavity closing to the endometrium, poor internal penetration, and no blood flow signal around the anechoic zone shown by color Doppler flow imaging (CDFI). These cystic anechoic zones can be divided into regular and irregular cystic zones [14]. The fluid accumulation in cystic anechoic zones is different from that in endometrial cavity fluid (ECF) and cesarean section scar diverticulum. ECF is often characterized by intimal line separation. Cesarean section scar diverticulum is a kind of uterine muscle wall cavity-like structure, which is related to the thinning of the uterine scar.

It has been widely recognized that such cystic anechoic zones in the uterine cavity may impact on pregnancy outcomes. In the existing literature, there is a limited understanding of this ultrasound phenomenon. It is speculated that it is associated with uterine lesions, which cause cystic changes in the endometrium and fluid accumulation in the uterine cavity [15].

At present, there is no relevant literature to study the effect of uterine cystic anechoic zones on pregnancy outcomes. In view of this, patients with such a condition were often faced with a dilemma: whether to proceed with fresh embryo transfer (ET) or cancel embryo transfer for a frozen cycle. Thus, the aim of this study is to investigate the effect of a newly observed cystic anechoic zone of uterine cavity during COH on the pregnancy outcomes of fresh-ET.

## 2. Materials and Methods

### 2.1. Study Design and Participants

A retrospective cohort study was conducted at the Center for Reproductive Medicine, Shandong University, Jinan, China. This study analyzed data from patients who underwent IVF/ICSI between 1 January 2014 and 31 December 2020. During COH, a transvaginal ultrasound (TVU) was performed to assess follicular development and endometrium status up to the last visit before ET. The cystic anechoic zone of the uterine cavity occurred in the last ultrasound before the injection of human chorionic gonadotropin (hCG) during COH, which was the focus of the study.

The inclusion criteria were as follows: (1) female age, 20–45 years; (2) performing fresh-ET; and (3) complete clinical information and follow-up information. This study did not restrict the number of IVF/ICSI cycles for patients. Exclusion criteria were: (1) incorrect clinical data; (2) oocyte donation cycles; (3) not the first cycle to discover a cystic anechoic zone of uterine cavity; (4) abnormal pregnancy history; (5) uterine malformations; (6) disappeared cystic anechoic zone of uterine cavity before fresh-ET; (7) patients with untreated uterine neoplasm, submucosal uterine fibroids or EP; (8) patients with a cystic anechoic zone of uterine cavity before COH; (9) patients with ECF during COH; (10) patients with hydrosalpinx; (11) patients with cesarean section scar diverticulum; or (12) the cystic anechoic zone between the muscular layers of the uterus. It is worth mentioning that donor semen could be used as there was no male factor rejection.

### 2.2. IVF/ICSI-ET Procedures

Ovarian stimulation protocol, oocyte retrieval, fertilization and fresh-ET were performed according to standard procedures used by our hospital [16]. The COH protocol for each patient was set according to the patient’s condition. Briefly, the follicular growth was monitored by assay of E2 levels as well as TVU. Next, when the diameters of two or more follicles were measured at ≥ 18 mm, a 4000–10,000 IU dose of hCG was injected intramuscularly. Then, oocyte retrieval was performed 34–36 h later. Women whose partners had poor sperm quality were inseminated with ICSI procedures, while conventional IVF was used for other patients approximately 4–6 h after follicle aspiration. Morphologic criteria were used for embryo scoring. The system of embryo morphology scoring was based on the scoring of cell number, fragmentation and symmetry for early-stage embryos and expansion, inner cell mass (ICM) and trophectoderm for blastocysts [17]. All patients included in the study had good/fair-quality embryos. Two fresh D3 embryos were routinely transferred in our center. For patients with special conditions, such as uterine anomalies, scarred uterus or those who wish to have a singleton pregnancy, single blastocyst transfer was recommended to reduce the risks of multiple pregnancies.

### 2.3. Ultrasonography Examination

A two-dimensional transvaginal sonography scan with Samsung HS50 (Samsung Medison Co. Ltd., Gang-won-do, Korea) was performed as a routine step to assess follicular development and endometrium condition during COH. The endometrium was scanned sagittally along the mid-line axis of the uterus. According to the patient’s COH protocol, multiple transvaginal ultrasound examinations were performed at different periods in the menstrual cycles. Measuring the cystic anechoic zone diameter between endometrial linings in a sagittal view is part of our routine ultrasound measurement that needs to be documented and the average diameter should be used as an index for analysis. In addition, the type of endometrium on the day of hCG injection was recorded. The endometrium can be divided into type A, type B and type C. Type A is defined as a completely uniform, hyperechoic endometrium. Type B is defined as an intermediate type, which is characterized by the same ultrasonic reflectivity as that of the myometrium, with no obvious echo line or no echo line in the center. Type C is defined as a multi-layered endometrium consisting of prominent external and midline high echo lines and internal hypoechoic areas [18]. The quality control team of the ultrasound department is responsible for the quality management of the results.

### 2.4. Outcome Measures

Live birth was the primary outcome that was defined as delivery of any viable neonate at 28 weeks or more of gestation. Other outcomes mainly included biochemical pregnancy, clinical pregnancy and pregnancy loss (miscarriage during biochemical pregnancies and clinical pregnancies). Biochemical pregnancy was usually defined using β-hCG levels above 10 IU/L. It was measured at 14 or 12 days after transfer of day-3 embryos or day-5 blastocyst transfers, respectively. Clinical pregnancy was defined as the presence of a gestational sac in the uterine cavity at 35 days after embryo transfer, as detected on ultrasonography [16]. Biochemical pregnancy loss was defined as spontaneous pregnancy demise based on a previous positive pregnancy test that then becomes negative without an ultrasound evaluation. Clinical pregnancy loss was defined as loss of a pregnancy after it has been identified on TVU [19].

### 2.5. Statistical Analysis

IBM SPSS Statistics (v. 26.0; International Business Machines Co., Armonk, NY, USA) was used for data analysis. Continuous data were expressed as mean ± standard deviation (SD) and assessed by Student’s *t*-test. Categorical data were expressed as frequencies and percentages, and differences between groups were tested by Chi-squared (χ^2^) test. A two-sided *p* < 0.05 was considered statistically significant in all analyses.

By using propensity score matching (PSM), confounding and selection bias were minimized [20]. In this study, variables that may be related to the pregnancy outcomes were selected as matching variables [21]. The variables included female age, body mass index (BMI), basal follicle-stimulating hormone (FSH), basal luteinizing hormone (LH), basal estradiol (E2), antral follicle count (AFC), type of infertility, indications for IVF/ICSI, protocol of COH, duration of ovarian stimulation, starting dosage of gonadotropin-releasing hormone (GnRH), total gonadotropin dose, E2 level on hCG trigger day, endometrial thickness (EMT) on hCG trigger day, number of oocytes retrieved and number of embryos transferred. Until the matching was completed, the researchers did not know how the pregnancy outcomes would turn out.

The PSM procedure was implemented using the SPSS PSM extension. To optimize the precision of the study, we performed one-to-two no-substitution matching based on the closest propensity scores between the two groups with a caliper width of 0.01. We calculated standardized differences (D) in all variables in propensity scores before and after matching to assess the effect of matching on disequilibrium. D < 0.1 was used as a threshold indicating negligible differences in the mean or prevalence of covariates between exposure groups.

To further verify the results, modified Poisson regression models were conducted using after-matching data, and RR and 95% confidence intervals (CI) before and after adjusting for confounders were calculated. Correlation analysis was used to explore whether there was a correlation between the two variables, and to provide a correlation coefficient (r) to analyze the correlation.

## 3. Results

A total of 88,518 patients were included in the study (Figure 1). After processing the data according to the exclusion criteria, the cystic anechoic zone group and no cystic anechoic zone group consisted of 182 and 30,531 patients, respectively. After matching, the cystic anechoic zone group finally included 179 patients, and the remaining group included 358 patients. Figure 2 shows the ultrasound appearance of classical cystic anechoic zones of the uterine cavity. However, the specific cause of the cystic anechoic zone could not be distinguished from the image.

Prior to PSM, the baseline characteristics and ET variables of the two cohorts were unevenly distributed. The patients in the cystic anechoic zone group had a longer time of ovarian stimulation (11.23 ± 2.88 vs. 10.37 ± 3.16, D = 0.299), more total gonadotropin dose (2412.43 ± 1213.42 vs. 2105.71 ± 1011.26, D = 0.253) and thicker EMT on hCG trigger day (1.14 ± 0.24 vs. 1.07 ± 0.21, D = 0.279). The PSM balanced these characteristics between the cohorts (Table 1), indicating that the matched cohorts had highly similar characteristics.

Comparison of pregnancy outcomes after PSM were listed in Table 2. Overall, there were significant differences in the incidence of live birth (38.0% vs. 48.6%, *p* = 0.025) and pregnancy loss during clinical pregnancy (22.2% vs. 12.4%, *p* = 0.031) in the cystic anechoic zone group as compared with the no cystic anechoic zone group. No differences were observed in the biochemical pregnancy rates with 57.5% in the cystic anechoic zone cohort versus 63.1% in the cohort without cystic anechoic zone (*p* = 0.210). The clinical pregnancy rates of the two groups were 50.3% and 56.4%, respectively, and there was no significant difference between them (*p* = 0.188). The rates of biochemical pregnancy loss were also similar with 12.6% versus 10.6% in these two cohorts, respectively (*p* = 0.593).

Modified Poisson regression models further demonstrated that the cystic anechoic zones had a significant adverse effect on live birth rate and clinical pregnancy miscarriage rate. As seen in Table 3, compared to the cystic anechoic zone group, the aRR for live birth rate and clinical pregnancy miscarriage rate in the no cystic anechoic zone cohort were 0.788 (95% CI, 0.638 to 0.974) and 0.557 (95% CI, 0.325 to 0.953), both of which were statistically significant (Table 3).

One patient in the cystic anechoic group had type C endometrium. The patient was not included in the analysis of uterine cystic anechoic zone and endometrial pattern. Regarding endometrial pattern, the proportion of type B endometrium in patients with a cystic anechoic zone is higher than that of type A endometrium (56.2% vs. 43.8%), while the proportion of endometrial types in the non-cystic anechoic zone group was 89.9% and 10.1% respectively, and the difference was statistically significant (*p* < 0.001) (Table 4).

The average diameter of multiple punctate cystic anechoic zones was about 0.1 cm after measurement. The diameters of two or more anechoic zones in the uterine cavity were recorded as the total average diameter after addition. The results showed no statistical difference in the association between live birth rate (r = −0.025, *p* = 0.742), clinical pregnancy rate (r = −0.032, *p* = 0.670) or biochemical pregnancy rate (r = −0.007, *p* = 0.921) and the diameter of the cystic anechoic zone in the uterine cavity (Table 5).

## 4. Discussion

To the best of our knowledge, this is the first cohort study investigating the pregnancy outcomes following fresh-ET of patients with or without cystic anechoic zones of uterine cavity. We found that the presence of a cystic anechoic zone was associated with prominently increased clinical pregnancy miscarriage rate and decreased live birth rate. The incidences of clinical pregnancy rate, biochemical pregnancy rate and biochemical pregnancy miscarriage rate were similar between the two groups. There was a significant difference in endometrial patterns (type A and type B) between the two groups. Additionally, we found no correlation between the diameter of cystic anechoic zones and pregnancy outcomes.

In clinical work, we often confuse the cystic anechoic zone of uterine cavity with ECF. ECF consisted of blood, mucus, endometrial secretions and/or fallopian tube fluid [22]. Fluid accumulation within the uterine cavity was manifested as an echogenic transparent ring structure dilated by a certain amount of fluid, while the cystic anechoic zone of uterine cavity showed local fluid accumulation between the intima. The uterine cystic anechoic zone was not produced simply by the accumulation of fluid in the uterine cavity, but by various diseases which affect the endometrium [16].

Prior to COH, all enrolled patients underwent hysteroscopy to exclude uterine cavity lesions. After PSM, the thickness of the endometrium in the cystic anechoic area group was 1.12 ± 0.23 cm, while that in the control group was 1.09 ± 0.20 cm. Since the EMT on hCG trigger day was matched by PSM, there was no significant difference in EMT between the two groups, but from the ultrasound results of patients with cystic anechoic zones, we can find that most patients have multiple punctate anechoic zones. This ultrasound manifestation was considered to be a manifestation of EH [16,23]. Meanwhile, as long as the thickness of the endometrium was more than 1.0 cm, EH can be considered [24]. What’s more, chronic endometritis (CE) can also cause cystic anechoic zones. CE was common at our center. It is reported that the prevalence of CE in premenopausal women ranged from 8% to 72% [25]. Ultrasound images showed normal morphology of the intima, uneven echo and irregular cystic anechoic zone. In addition, patients with uterine adhesions can also find cystic anechoic zones [26], mainly owing to the small cavity formed by the local adhesion of the endometrium, resulting in the inability of the intrauterine fluid to drain. The ultrasound showed one or more cystic zones in the uterine cavity disrupting the lining of the uterus. In cases of severe uterine adhesions, the uterine cavity may appear irregular with a loss of endometrial echo. In theory, uterine cystic anechoic zones may be caused by uterine adhesion, but the lesion was not found by hysteroscopy before COH so it was unlikely to be caused by uterine adhesion. What’s more, EP also did not seem to produce this ultrasound presentation due to the unlikely disappearance of the hyperechoic polyp component surrounding the cystic lesions of the polyps. Then, uterine adenomyosis, which grew and became more active during COH, produced cystic anechoic zones mostly in the endometrial–myometrial interface (EMI), which was inconsistent with the ultrasound presentation studied in this paper [27,28].

The present study showed that patients in the cystic anechoic zone cohort had a higher clinical pregnancy miscarriage rate and therefore a lower live birth rate than the control cohort. Some literature revealed that abnormal endometrial receptivity can lead to different pregnancy outcomes, such as complete implantation failure (infertility) and severe implantation defects (miscarriage) [29,30]. The factors (cytokines, growth factors, apoptotic proteins and so on) connected with endometrial receptivity were the main source of nutrients to support the development of the gestational body during the first trimester of pregnancy [31,32,33,34,35,36]. In the fresh ET cycle, the increase of endometrial thickness was related to the decrease of miscarriage rate. However, the live birth rate tended to stabilize after the endometrium reached 1.0–1.2 cm [37]. In this study, the average endometrial thickness of the two groups was similar, excluding the effect of endometrial thickness on pregnancy outcomes. The ways that CE affected pregnancy outcomes primarily included altering the expression related to endometrial receptivity [38]. More and more evidence showed that CE was not conducive to gestational placentation, which should be the potential cause of pregnancy loss [39,40]. In addition, patients with uterine adhesions had high miscarriage frequency because of having poor endometrial receptivity [41,42,43,44,45,46]. This lesion may lead to pregnancy loss due to insufficient endometrial development to support feto-placental growth [47]. The adverse effects of uterine adhesions on pregnancy outcomes may also be interrelated to the fluid in anechoic zones, which was similar to ECF. Some studies suggested that ECF can affect reproductive outcomes by reducting in endometrial receptivity [48,49,50,51,52]. To sum up, we inferred that a cystic anechoic zone may increase clinical pregnancy miscarriage rate by impairing endometrial receptivity and impeding embryo development.

Measuring the type of endometrium during the IVF cycle was a routine method to evaluate endometrial receptivity [53]. We found that there was a significant difference in the type of endometrium between the two groups (*p* < 0.001). This result may be due to the cystic anechoic zone affecting the state of the endometrium. Several studies had concluded that the live birth rate was significantly increased in patients with triple-line pattern (type A) endometrium on the day of administration of hCG [54,55,56,57], which was consistent with the results of this study. This further confirmed that cystic anechoic zones may lead to poor endometrial receptivity.

There was no correlation between the diameter of the cystic anechoic zone and the pregnancy outcomes. CE and EH were lesions that occur throughout the entire endometrium, and the cystic anechoic zones of the uterine cavity with different diameters were only the ultrasound manifestations of these lesions. Their impact on pregnancy outcomes was still through the lesions themselves. If fluid in cystic anechoic zones of patients with uterine adhesion was produced by reproductive tract physiologically, then the pregnancy rates will not decrease [58,59]. Since patients with hydrosalpinx were excluded in this study, the diameter of the cystic anechoic zone caused by fluid volume had no effect on the pregnancy rate.

There are a number of strengths associated with the current study. First, our study extensively controls for potential confounding differences between the group with and without cystic anechoic zones by PSM, as well as for intrinsic control of confounding factors. Second, our study can provide guidance to patients on whether to undergo fresh embryo transfer in the current cycle. However, there are some limitations of this study. First, this was a single-center retrospective study and the findings need to be further validated in a multicenter randomized controlled study. Secondly, endometrial receptivity indicators (endometrial thickness, etc.) are not analyzed. Third, without additional uterine cavity operation, it is impossible to further judge the true situation of the anechoic area. More detailed hysteroscopy can be performed when the cystic anechoic zone appears, such as hysteroscopy guided by ultrasound, and additional surgical instruments can be used to cut open the cystic anechoic zone and biopsy.

## 5. Conclusions

In conclusion, the cystic anechoic zone detected by an ultrasound during COH can significantly affect live birth rate and clinical pregnancy miscarriage rate. According to the ultrasonographic findings and pregnancy outcomes, it was inferred that the cystic anechoic zone of the uterine cavity caused by CE, uterine adhesion and EH may be related to impaired endometrial receptivity. We suggest that patients with cystic anechoic zones in the uterine cavity during COH should consider canceling fresh ET and transferring them in a subsequent cycle after the evaluation of the endometrium.

## Figures and Tables

**Figure 1 jcm-12-00134-f001:**
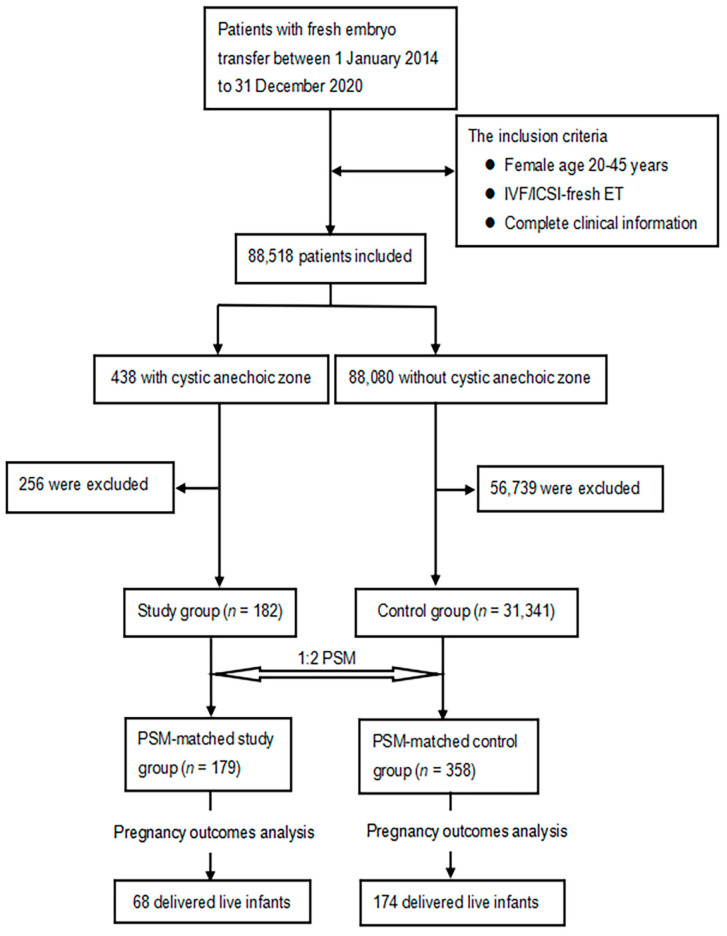
Flow chart of case screening and grouping in study group and control group.

**Figure 2 jcm-12-00134-f002:**
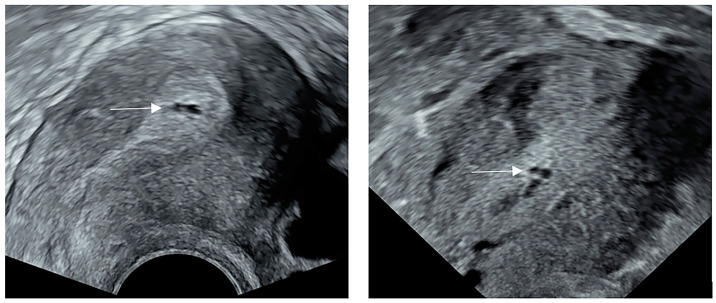
The ultrasound appearance of classical cystic anechoic zones of the uterine cavity. The white arrows show the location of the cystic anechoic zones of the uterine cavity.

**Table 1 jcm-12-00134-t001:** Baseline characteristics and embryo transfer variables of study patients before and after PSM.

Characteristic	Before PSM (*n* = 30,713)			After PSM (*n* = 537)		
	Cystic Anechoic Zone	No Cystic Anechoic Zone	*D	Cystic Anechoic Zone	No Cystic Anechoic Zone	*D
	(*n* = 182)	(*n* = 30,531)		(*n* = 179)	(*n* = 358)	
Age (years)	32.65 ± 4.67	32.29 ± 4.97	0.077	32.93 ± 4.66	33.21 ± 5.70	−0.030
BMI (kg/m^2^)	24.43 ± 3.69	23.89 ± 3.63	0.146	24.45 ± 3.63	24.68 ± 4.10	−0.024
Baseline sex hormone						
Basal FSH (IU/L)	7.03 ± 3.31	7.33 ± 6.42	−0.090	7.13 ± 3.38	7.09 ± 3.15	0.062
Basal LH (IU/L)	5.19 ± 5.41	5.64 ± 8.21	−0.083	5.12 ± 5.14	5.42 ± 2.43	0.074
Basal E2 (pg/mL)	52.21 ± 131.31	78.41 ± 216.43	−0.200	50.73 ± 123.99	57.32 ± 134.99	−0.030
AFC, no.						
Left ovary	6.42 ± 3.93	6.07 ± 4.17	0.081	6.36 ± 3.87	6.55 ± 4.16	−0.049
Right ovary	6.53 ± 3.75	6.25 ± 4.20	0.075	6.42 ± 3.66	6.28 ± 4.67	0.039
Type of infertility, no. (%)			0.100			−0.011
Primary infertility	74 (40.7)	14,281 (45.6)		75 (36.6)	60 (29.3)	
Secondary infertility	108 (59.3)	17,060 (54.4)		130 (63.4)	145 (70.7)	
Indications for IVF/ICSI, no. (%)			0.049			0.031
Male factor	25 (13.7)	7530 (24.0)		26 (12.7)	37 (18.0)	
Tubal factor	139 (76.4)	19,985 (63.8)		157 (76.6)	142 (69.3)	
Combined factors	9 (4.9)	740 (2.4)		9 (4.4)	6 (2.9)	
Others	9 (4.9)	3086 (9.8)		13 (6.3)	20 (9.8)	
Protocol of COH, no. (%)			0.248			−0.011
Short agonist	36 (19.8)	8214 (26.2)		43 (21.0)	42 (20.5)	
Long agonist	66 (36.3)	13,255 (42.3)		73 (35.6)	86 (42.0)	
Others	80 (44.0)	9872 (31.5)		89(43.4)	77 (37.6)	
Duration of ovarian stimulation (days)	11.23 ± 2.88	10.37 ± 3.16	0.299	11.14 ± 2.83	10.97 ± 2.55	0.049
Starting dosage of GnRH (IU)	179.90 ± 54.22	187.26 ± 64.09	−0.136	183.62 ± 57.29	184.76 ± 64.89	−0.020
Total dosage of GnRH (IU)	2412.43 ± 1213.42	2105.71 ± 1011.26	0.253	2418.48 ± 1200.73	2343.05 ± 1188.46	0.065
E2 level on HCG trigger day (pg/mL)	2702.25 ± 1546.78	2743.31 ± 1494.77	−0.027	2570.42 ± 1653.43	2560.72 ± 1715.50	−0.025
EMT on hCG trigger day (cm)	1.14 ± 0.24	1.07 ± 0.21	0.279	1.12 ± 0.23	1.09 ± 0.20	−0.066
Number of oocytes retrieved, no.	8.76 ± 4.53	8.88 ± 4.51	−0.026	8.76 ± 4.58	8.57 ± 4.53	−0.003
Number of embryos transferred, no.	1.63 ± 0.52	1.71 ± 0.51	−0.161	1.59 ± 0.52	1.56 ± 0.50	−0.011

Values are presented as mean ± standard deviation or no. (%). PSM, propensity score matching; BMI, body mass index; FSH, follicle stimulating hormone; LH, luteinizing hormone; E2, estradiol; AFC, antral follicle count; IVF, in vitro fertilization; ICSI, intracytoplasmic sperm injection; COH, controlled ovarian hyperstimulation; GnRH, gonadotropin-releasing hormone; hCG, human chorionic gonadotropin; EMT, endometrial thickness. *D: Standardized difference. D ≥ 0.1 are printed in bold. D less than 0.1, cohorts can be considered to be balanced with respect to the demographics assessed.

**Table 2 jcm-12-00134-t002:** Pregnancy outcomes of the participants before and after PSM.

Outcomes	Before Matching (*n* = 31,523)			After Matching (*n* = 537)		
	Cystic Anechoic Zone (*n* = 182)	No Cystic Anechoic Zone (*n* = 31,341)	*p*	Cystic Anechoic Zone (*n* = 179)	No Cystic Anechoic Zone (*n* = 358)	*p*
Biochemical pregnancy rate, % (no.)	57.7 (105)	60.7 (19,038)	0.400	57.5 (103)	63.1 (226)	0.210
Clinical pregnancy rate, % (no.)	50.0 (91)	53.4 (16,726)	0.364	50.3 (90)	56.4 (202)	0.188
Live birth rate, % (no.)	37.9 (69)	43.5 (13,631)	0.130	38.0 (68)	48.6 (174)	0.025
Pregnancy loss, % (no./total no.)						
During biochemical pregnancy	13.3 (14/105)	12.1 (2312/19,038)	0.710	12.6 (13/103)	10.6 (24/226)	0.593
During clinical pregnancy	22.0 (20/91)	16.5 (2763/16,726)	0.162	22.2 (20/90)	12.4 (25/202)	0.031

*p* > 0.05, not significant.

**Table 3 jcm-12-00134-t003:** Crude and adjusted RR of cystic anechoic zone against no cystic anechoic zone for pregnancy outcomes.

Outcomes	Crude RR (95%CI)	*p*	* Adjusted RR (95%CI)	*p*
Biochemical pregnancy rate	0.912 (0.786–1.058)	0.222	0.909 (0.786–1.051)	0.198
Clinical pregnancy rate	0.891 (0.750–1.058)	0.188	0.887 (0.748–1.051)	0.166
Live birth rate	0.782 (0.630–0.969)	0.025	0.788 (0.638–0.974)	0.027
Pregnancy loss				
During biochemical pregnancy	0.841 (0.447–1.585)	0.593	0.839 (0.441–1.597)	0.593
During clinical pregnancy	0.557 (0.327–0.949)	0.031	0.557 (0.325–0.953)	0.033

RR, relative risk; CI, confidence interval; *p* > 0.05, not significant. * Adjusted for age, BMI, basal FSH, basal LH, basal E2, starting dosage of GnRH (IU), total dosage of GnRH (IU) and protocol of COH.

**Table 4 jcm-12-00134-t004:** Percentage of endometrium patterns in study and control groups.

	A Type (*n* = 400)	B Type (*n* = 136)	*p*
Cystic anechoic zone	78 (43.8%)	100 (56.2%)	<0.001
No cystic anechoic zone	322 (89.9%)	36 (10.1%)	

The endometrium pattern was measured on hCG trigger day.

**Table 5 jcm-12-00134-t005:** Correlation between the diameter of uterine cystic anechoic zone and pregnancy outcomes.

	The Diameter of Uterine Cystic Anechoic Zone (cm)			r	*p*
	0.1 < d< 0.3 (*n* = 106)	0.3 ≤ d < 0.6 (*n* = 49)	0.6 ≤ d ≤ 1.8 (*n* = 24)		
Biochemical pregnancy rate, % (no.)	57.5 (61)	59.2 (29)	54.2 (13)	−0.007	0.921
Clinical pregnancy rate, % (no.)	50.9 (54)	53.1 (26)	41.7 (10)	−0.032	0.670
Live birth rate, % (no.)	38.7 (41)	38.8 (19)	33.3 (8)	−0.025	0.742

d, diameter; r, correlation coefficient; *p* > 0.05, not significant.

## Data Availability

Not applicable.

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
