# Peer review of "The Cystic Anechoic Zone of Uterine Cavity Newly Observed during Controlled Ovarian Hyperstimulation Affects Pregnancy Outcomes of Fresh Embryo Transfer"

_jcm, 2022, doi:10.3390/jcm12010134_

Round 1
Reviewer 1 Report
Well qualified article. I read with great interest.
According to this paper the cystic anechoic zone detected by ultrasound during COH was statistically associated with live birth rate and clinical pregnancy miscarriage rate. According to the ultrasonographic findings and pregnancy outcomes, it was inferred that the cystic anechoic zone of uterine cavity caused by CE, uterine adhesion and EH may be related to endometrial receptivity damage.The authors suggest that patients with cystic anechoic zones in the uterine cavity during COH should consider canceling fresh ET and continuing IVF/ISCI after more accurate examination.
Author Response
Dear Reviewer:
On behalf of all the contributing authors, I would like to express our sincere appreciation of the reviewer’s constructive comments concerning our article entitled “The cystic anechoic zone of uterine cavity newly observed during controlled ovarian hyperstimulation affects pregnancy outcomes of fresh embryo transfer” (Manuscript ID: jcm-2104863). These comments are all valuable for our article.
We appreciate for Reviewer’s warm work earnestly. Once again, thank you very much for your comments and suggestions.
Reviewer 2 Report
Thank you for the opportunity to review this manuscript titled “The cystic anechoic zone of uterine cavity newly observed during controlled ovarian hyperstimulation affects pregnancy outcomes of fresh embryo transfer” by Yizheng Tian et al.
The authors have studied a valid research question. They are coming up with a new method of deciding the suitability of embryo transfer in the same cycle. They have a sound study design and analysis from which they have come up with the conclusion that the live birth rate of patients with uterine cystic anechoic zones was lower than that in the no uterine cystic anechoic zone group. I have no major comments.
I put forward the following minor comments.
Introduction
Line 32-37 – It is inappropriate to add a categorization of endometrium in the first paragraph. Rather a first paragraph should focus on drawing the attention of the reader to the clinical problem you are trying to find answers for.
Materials and methods
Line 91 – “growth of follicular was” should be corrected as “growth of follicles were” or “follicular growth was”
Line 96 – Please state the method of morphological criteria used for embryo assessment.
Line 98 – Since the investigators have analyzed the embryos of 2 stages (day 3 and 5), state how the differences of that may have affected the results.
Line 108 – Remove the full stop after “measurement”
Line 109 – The type of endometrium classification, mentioned in the first paragraph, may be inserted here.
Results
The number of patients included in the study has been mentioned as 88518. Isn’t it the number of patients screened, rather than the sample the number (n)?
Figure 1 has incorrect capitalization of words. Ex. 88518 Patients included.
Figure 2: What is indicated by the arrow? Please mention it in the figure legend.
Line 171: “Table â… ” should be corrected as Table 1. Same for other table numbers.
Table 2: When you provide a rate, invariably you calculate a percentage. Here you are providing numbers as well.
Ex. “Biochemical pregnancy rate, no. (%)”: Here authors mention a rate. However, the main information they provide is a number and within brackets a percentage is given. The percentage can be the rate. In that case, why it is mentioned within brackets?
“One patient in the cystic anechoic group had type C endometrium. The patient was not included in the analysis of uterine cystic anechoic zone and endometrial pattern”. If the presence of type C endometrium is a reason for not including in the analysis, why the authors did not consider it as an exclusion criterion?
Line 211: The diameter should be corrected as “The average diameter”.
Table 5: “The pregnancy outcomes of different diameters of uterine cystic anechoic zone”. The table heading is not clear. Please reword it.
Discussion
The discussion is well written.
However, it is surprising to see the 1st paragraph in the discussion. Surely, this is a previous text about what to be included in the discussion. It is doubtful if the co-authors have read the manuscript. Otherwise, how come this was missed by all the authors? Please remove it.
Line 284-285 – Does it mean impaired endometrial receptivity includes miscarriage after implantation?
Line 310 : Use past tense.
Conclusion
Line 317 : “ statistically associated with live birth rate and clinical pregnancy miscarriage rate”. Please amend this. “Statistically associated with” is not appropriate wording.
Since this section is about the conclusion, there is no need to start the conclusion as “in conclusion”.
Line 320: “related to endometrial receptivity damage” – may change to “related to impaired endometrial receptivity”.
Line 322: “canceling fresh ET and continuing IVF/ISCI after more accurate examination”. This is confusing. If you cancel fresh embryo transfer, the next option is frozen-thawed embryo transfer in a subsequent cycle. Stating “continuing IVF/ISCI after more accurate examination” could be replaced with an alternative statement. For example, canceling fresh ET and transferring them in a subsequent cycle after the evaluation of the endometrium.
Reviewer 3 Report
Thank you for requesting to provide a review of this article, which has a subject of high interest.
The main purpose of the analysis was to investigate the effect of a newly observed cystic anechoic zone of uterine cavity during COH on the pregnancy outcomes of fresh - E5. The conducted study was a retrospective cohort analysis , between January 2014 and December 2020 and followed a group of 88518 patients.
Regarding the structure and accuracy of the phrases, the manuscript has well structured information, with supported evidence and well structured phrases.
The manuscript is original and well defined. The results provide an advance in current knowledge. The results are being interpreted appropriately and are significant, as well as the conclusions.
The article is written in an appropriate way.
The study is correctly designed and the analysis is being performed at high standards, so the data are robust enough to draw the conclusion.
Surely the paper will attract a wide readership.
The English language is appropriate and well understandable.
There are a few things to add in the lines below, but the article should be published after the corrections are made:
Line 39: that appeared, not „appeared”
Line 60: „,” before „causing”
Line 76: „,” before „was the focus of the study”
Line 107: without „.” between „and” and „needed”
